# Neural Ensemble Search via Bayesian Sampling

Yao Shu[1]      Yizhou Chen[1]      Zhongxiang Dai[1]      Bryan Kian Hsiang Low[1]

[1]Department of Computer Science, National University of Singapore, Singapore

## Abstract

Recently, *neural architecture search* (NAS) has been applied to automate the design of neural networks in real-world applications. A large number of algorithms have been developed to improve the search cost or the performance of the final selected architectures in NAS. Unfortunately, these NAS algorithms aim to select only *one single* well-performing architecture from their search spaces and thus have overlooked the capability of *neural network ensemble* (i.e., an ensemble of neural networks with diverse architectures) in achieving improved performance over a single final selected architecture. To this end, we introduce a novel neural ensemble search algorithm, called *neural ensemble search via Bayesian sampling* (NESBS), to effectively and efficiently select well-performing neural network ensembles from a NAS search space. In our extensive experiments, NESBS algorithm is shown to be able to achieve improved performance over state-of-the-art NAS algorithms while incurring a comparable search cost, thus indicating the superior performance of our NESBS algorithm over these NAS algorithms in practice.

## 1   INTRODUCTION

Recent years have witnessed a surging interest in designing well-performing architectures for different tasks. These architectures are typically manually designed by human experts, which requires numerous trials and errors during this manual design process and therefore is prohibitively costly. Consequently, the increasing demand for developing well-performing architectures in different tasks makes this manual design infeasible. To avoid such human efforts, Zoph and Le [2017] have introduced *neural architecture search* (NAS) to help automate the design of architectures.

Since then, a number of NAS algorithms [Pham et al., 2018, Liu et al., 2019, Chen et al., 2019] have been developed to improve the search efficiency (i.e., search cost) or the search effectiveness (i.e., generalization performance of their final selected architectures) in NAS.

However, conventional NAS algorithms aim to select only *one single architecture* from their search spaces and have thus overlooked the capability of other candidate architectures from the same search spaces in helping improve the performance achieved by their final selected single architecture. That is, *neural network ensembles* are widely known to be capable of achieving an improved performance compared with a single neural network in practice [Cortes et al., 2017, Gal and Ghahramani, 2016, Lakshminarayanan et al., 2017]. This naturally begs the question: *How to select best-performing neural network ensembles with diverse architectures from a NAS search space in order to improve the performances achieved by existing NAS algorithms?* To the best of our knowledge, only limited efforts (e.g., [Zaidi et al., 2021]) have been devoted to this problem in the NAS literature. Unfortunately, the *neural ensemble search* (NES) algorithm based on random search or evolutionary algorithm in [Zaidi et al., 2021] requires excessive search costs to select their final neural network ensembles, which will not be affordable in resource-constrained scenarios.

To this end, this paper introduces a novel algorithm, namely *neural ensemble search via Bayesian sampling* (NESBS), to effectively and efficiently select the well-performing neural network ensemble with diverse architectures from a search space. We firstly represent the search space as a supernet following conventional one-shot NAS algorithms and then use the model parameters inherited from this supernet after its model training to estimate the single-model performances and also the ensemble performance of independently trained architectures (Sec. 3.1). Next, since both single-model performances and diverse model predictions affect the final ensemble performance according to [Zhou, 2012], we propose to use a variational posterior distribution of architectures based on a trained supernet to characterize these two factors,

*Accepted for the 38th Conference on Uncertainty in Artificial Intelligence* (UAI 2022).

i.e., single-model performances and diverse model predictions (Sec. 3.2). We then introduce two novel Bayesian sampling algorithms based on the posterior distribution of architectures, i.e., *Monte Carlo sampling* (MC Sampling) and *Stein Variational Gradient Descent with regularized diversity* (SVGD-RD), to effectively and efficiently select ensembles with both competitive single-model performances and compelling diverse model predictions (Sec. 3.3), which is also guaranteed to be able to achieve impressive ensemble performances [Zhou, 2012]. Lastly, we use extensive experiments to show that our NESBS algorithm is indeed able to select well-performing neural network ensembles effectively and efficiently in practice (Sec. 4).

# 2   RELATED WORKS & BACKGROUND

## 2.1   NEURAL ARCHITECTURE SEARCH

In the literature, many NAS algorithms [Real et al., 2019, Zoph and Le, 2017, Zoph et al., 2018] have been developed to automate the design of well-performing neural architectures. However, these NAS algorithms are inefficient in practice due to their requirement of the independent model training for each candidate architecture in the search space. To reduce such training costs, a supernet has been introduced to represent the search space and also share model parameters among the candidate architectures in the search space [Pham et al., 2018]. As a result, only the model training of this supernet is required, which can significantly improve the search efficiency of conventional NAS algorithms. After that, a number of one-shot NAS algorithms based on model parameter sharing [Chen et al., 2019, Chen and Hsieh, 2020, Chu et al., 2020, Liu et al., 2019, Xie et al., 2019] have been developed. Unfortunately, these algorithms aim to select *only one single architecture* from their search spaces. Thus, the capability of other candidate architectures from the same search spaces in helping improve the performance of their final selected single architecture have been overlooked.

## 2.2   NEURAL NETWORK ENSEMBLES

Meanwhile, neural network ensembles have been widely applied to improve the performance of a single neural network in different applications [Dietterich, 2000]. Over the years, a number of methods have been proposed to construct such neural network ensembles. For example, Gal and Ghahramani [2016] have proposed to use Monte Carlo Dropout to obtain neural network ensembles at test time. Meanwhile, *deep ensembles* (DeepEns) [Lakshminarayanan et al., 2017] adopt neural networks trained with different random initializations to construct ensembles and has achieved impressive performances on various tasks. Another line of ensemble works uses the checkpoints obtained during model training to build neural network ensembles [Huang

et al., 2017]. More recently, Zaidi et al. [2021] have introduced *neural ensemble search* (NES) into NAS area to build well-performing neural network ensembles by selecting diverse architectures from the NAS search space, which has achieved competitive performance even compared with other ensemble methods. Unfortunately, the algorithm presented in [Zaidi et al., 2021] is shown to be prohibitively costly, which will not be affordable in resource-constrained scenarios. To this end, this paper presents a novel NESBS algorithm to advance this line of works (e.g., NES) by achieving state-of-the-art performances for neural network ensembles with diverse architectures while incurring a reduced search cost.

## 2.3   STEIN VARIATIONAL GRADIENT DESCENT

*Stein Variational Gradient Descent* (SVGD) [Liu and Wang, 2016] is a variational inference algorithm that approximates a target distribution $p(\boldsymbol{x})$ with a simpler density $q^*(\boldsymbol{x})$ in a predefined set $\mathcal{Q}$ by minimizing the *Kullback-Leibler* (KL) divergence between these two densities:

$$q^* = \arg\min_{q \in \mathcal{Q}}\{\mathrm{KL}(q||p) \triangleq \mathbb{E}_q\left[\log\left(q(\boldsymbol{x})/p(\boldsymbol{x})\right)\right]\} . \quad (1)$$

Specifically, SVGD represents $q^*(\boldsymbol{x})$ with a set of particles $\{\boldsymbol{x}_i\}_{i=1}^n$ which are firstly randomly initialized and then iteratively updated with updates $\boldsymbol{\phi}^*(\boldsymbol{x}_i)$ and a step size $\epsilon$:

$$\boldsymbol{x}_i \leftarrow \boldsymbol{x}_i + \epsilon\boldsymbol{\phi}^*(\boldsymbol{x}_i) \quad \text{for } i = 1, \ldots, n . \quad (2)$$

Let $q_{[\epsilon\phi]}$ denote the distribution of updated particles $\boldsymbol{x}' = \boldsymbol{x} + \epsilon\boldsymbol{\phi}(\boldsymbol{x})$. Let $\mathbb{F}$ denote the unit ball of a vector-valued *reproducing kernel Hilbert space* (RKHS) $\mathcal{H} \triangleq \mathcal{H}_0 \times \ldots \times \mathcal{H}_0$ where $\mathcal{H}_0$ is an RKHS formed by scalar-valued functions associated with a positive definite kernel $k(\boldsymbol{x}, \boldsymbol{x}')$. The work of Liu and Wang [2016] has shown that (2) can be viewed as functional gradient descent in the RKHS $\mathcal{H}$ and the optimal $\boldsymbol{\phi}^*$ in (2) can be obtained by solving the following problem:

$$\boldsymbol{\phi}^* = \arg\max_{\boldsymbol{\phi}\in\mathbb{F}}\left\{-\frac{d}{d\epsilon}\mathrm{KL}(q_{[\epsilon\phi]}||p)\Big|_{\epsilon=0}\right\} , \quad (3)$$

which yields a closed-form solution:

$$\boldsymbol{\phi}^*(\cdot) = \mathbb{E}_{\boldsymbol{x}\sim q}[k(\boldsymbol{x},\cdot)\nabla_{\boldsymbol{x}}\log p(\boldsymbol{x}) + \nabla_{\boldsymbol{x}}k(\boldsymbol{x},\cdot)] . \quad (4)$$

In practice, Liu and Wang [2016] have approximated the expectation in this closed-form solution with the empirical mean of particles: $\boldsymbol{\phi}^*(\boldsymbol{x}_i) \approx \widehat{\boldsymbol{\phi}}^*(\boldsymbol{x}_i)$ where $\widehat{\boldsymbol{\phi}}^*(\boldsymbol{x}_i)$ is defined as

$$\widehat{\boldsymbol{\phi}}^*(\boldsymbol{x}_i) \triangleq \frac{1}{n}\sum_{j=1}^n k(\boldsymbol{x}_j, \boldsymbol{x}_i)\nabla_{\boldsymbol{x}_j}\log p(\boldsymbol{x}_j) + \nabla_{\boldsymbol{x}_j}k(\boldsymbol{x}_j, \boldsymbol{x}_i) .$$

$$(5)$$

As revealed in [Liu and Wang, 2016], the two terms in the aforementioned closed-form solution take different effects: The first term with $\nabla_{\boldsymbol{x}}\log p(\boldsymbol{x})$ favors particles with higher probability density, while the second term pushes the particles away from each other to encourage diversity.

# 3 NEURAL ENSEMBLE SEARCH VIA BAYESIAN SAMPLING

Contrary to the selection of one single architecture in conventional NAS algorithm, this paper focuses on the problem of selecting a well-performing neural network ensemble with diverse architectures from the NAS search space, i.e., *neural ensemble search* (NES) [Zaidi et al., 2021]. Let $f_{\mathcal{A}}(x, \theta_{\mathcal{A}})$ denote the output of an architecture $\mathcal{A}$ with input data $x$ and model parameter $\theta_{\mathcal{A}}$, $S$ be a set of architectures, $\Theta_S$ be a set of the corresponding model parameters of these architectures, and $\mathcal{L}_{\text{train}}$ and $\mathcal{L}_{\text{val}}$ denote the training and validation losses, respectively. Given the ensemble scheme $\mathcal{F}_S(x, \Theta_S) \triangleq n^{-1} \sum_{\mathcal{A} \in S} f_{\mathcal{A}}(x, \theta_{\mathcal{A}})$ with an ensemble size of $|S| = n$,[1] NES can be formally framed as

$$\min_S \mathcal{L}_{\text{val}}(\mathcal{F}_S(x, \Theta_S^*))$$
$$\text{s.t. } \forall \theta_{\mathcal{A}}^* \in \Theta_S^* \quad \theta_{\mathcal{A}}^* = \arg\min_{\theta_{\mathcal{A}}} \mathcal{L}_{\text{train}}(f_{\mathcal{A}}(x, \theta_{\mathcal{A}})) . \quad (6)$$

Unfortunately, (6) is challenging to solve mainly due to the following two reasons: (I) The enormous number of candidate architectures in the NAS search space (e.g., $\sim 10^{25}$ in the DARTS search space [Liu et al., 2019]) makes the independent model training of every candidate architecture (i.e., lower-level optimization in (6)) unaffordable. (II) The ensemble search space is exponentially increasing in the ensemble size $n$: For example, there are $\sim m^n$ different ensembles given $m$ diverse architectures. The combinatorial optimization problem (i.e., upper-level optimization in (6)) is thus intractable to solve within this huge ensemble search space. Recently, Zaidi et al. [2021] have attempted to avoid these two problems by sampling a small pool of architectures from the search space for their final ensemble search. Thus, they fail to explore the whole search space and may achieve poor ensemble performances in practice. Moreover, their search cost is still unaffordable due to the independent model training of every architecture in the pool.

To this end, we novelly present the *neural ensemble search via Bayesian sampling* (NESBS) algorithm to solve (6) effectively and efficiently. We firstly employ the model parameters inherited from a supernet (i.e., a representation of the NAS search space) after its model training to estimate the single-model performances and also the ensemble performance of independently trained architectures (Sec. 3.1). This only requires the model training of the supernet and thus allows us to overcome the aforementioned challenge I. We then derive a posterior distribution of architectures to characterize both the single-model performances and the diverse model predictions of candidate architectures in the search space (Sec. 3.2). Finally, based on this posterior distribution and also the aforementioned ensemble performance estimation, we introduce *Monte Carlo Sampling* (MC

[1]We apply this ensemble scheme for simplicity. Other ensemble schemes can also be used in the algorithm of this paper.

---

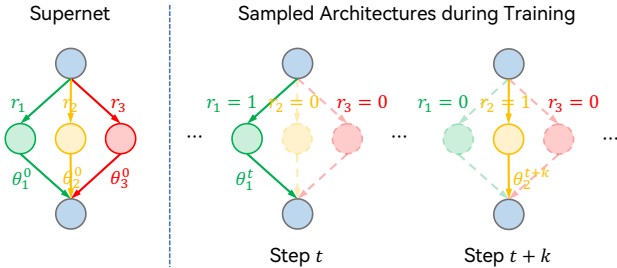

Figure 1: An illustration of the model training of supernet. The supernet here consists of three candidate architectures with $r_i$ indicating the selection of one architecture and $\theta_i^t$ denoting its model parameters at step $t$. In every training step, only one architecture is uniformly sampled to update its parameters and all other architectures will be ignored.

Sampling) and *Stein Variational Gradient Descent with regularized diversity* (SVGD-RD) to explore the ensembles in the whole search space effectively and efficiently (Sec. 3.3), which thus allows us to overcome the aforementioned challenge II. An overview of our NESBS is in Algorithm 1.

## 3.1 MODEL TRAINING OF SUPERNET

Similar to one-shot NAS algorithms [Liu et al., 2019, Pham et al., 2018], we represent NAS search space as a supernet. This then allows us to use the model parameters inherited from this trained supernet to estimate not only the single-model performances but also the ensemble performance of independently trained candidate architectures in the search space. However, in order to realize an accurate and fair estimation of these performances, we need to further ensure that every candidate architecture in the search space is trained for a comparable number of steps, namely, the training fairness among candidate architectures [Chu et al., 2019]. To achieve this, in every training step of this supernet, we uniformly randomly sample one single candidate architecture from this supernet for model training (see Fig. 1). The training fairness of such a training scheme can then be theoretically guaranteed, as demonstrated in Appendix A. Moreover, we provide empirical results in Appendix C.1 to validate the effectiveness of such performance estimations.

## 3.2 DISTRIBUTION OF ARCHITECTURES

It has been demonstrated that both competitive single-model performances and diverse model predictions are required to achieve compelling ensemble performances [Zhou, 2012]. That is, NES algorithms should be capable of selecting architectures with both competitive single-model performances and diverse model predictions to achieve competitive ensemble performances. To realize this, we introduce a posterior distribution of architectures to firstly characterize these two

factors. Let $\mathcal{D}$ denote the validation dataset, and $p(\mathcal{A})$ and $p(\mathcal{A}|\mathcal{D})$ denote, respectively, the prior and posterior distributions of a candidate architecture after its model training where $p(\mathcal{A})$ follows from a categorical uniform distribution, as required in Sec. 3.1. According to the Bayes' theorem, since $p(\mathcal{A})$ is uniform and $p(\mathcal{D})$ is constant,

$$p(\mathcal{A}|\mathcal{D}) = p(\mathcal{D}|\mathcal{A})p(\mathcal{A})/p(\mathcal{D}) \propto p(\mathcal{D}|\mathcal{A}) \quad (7)$$

where $p(\mathcal{D}|\mathcal{A})$ (i.e., likelihood) is widely used to represent the single-model performance (i.e., loss) in practice. So, (7) implies that the posterior distribution $p(\mathcal{A}|\mathcal{D})$ can also characterize the single-model performances of architectures.

Meanwhile, given a $\gamma$-Lipschitz continuous loss function $\mathcal{L}(\boldsymbol{f})$, the diversity of model predictions (i.e., $\|\boldsymbol{f}_{\mathcal{A}_1} - \boldsymbol{f}_{\mathcal{A}_2}\|_2$) can then be lower bounded based on the Lipschitz continuity of $\mathcal{L}(\cdot)$:

$$\|\boldsymbol{f}_{\mathcal{A}_1} - \boldsymbol{f}_{\mathcal{A}_2}\|_2 \geq \gamma^{-1}|\mathcal{L}(\boldsymbol{f}_{\mathcal{A}_1}) - \mathcal{L}(\boldsymbol{f}_{\mathcal{A}_2})| . \quad (8)$$

Therefore, (8) suggests that in addition to being able to characterize the single-model performances of architectures (i.e., $\mathcal{L}(\boldsymbol{f})$), the posterior distribution $p(\mathcal{A}|\mathcal{D})$ can estimate the diversity of model predictions for different architectures (e.g., $\mathcal{A}_1$ and $\mathcal{A}_2$) using $|p(\mathcal{A}_1|\mathcal{D}) - p(\mathcal{A}_2|\mathcal{D})|$.

However, it is intractable to obtain exact posterior distribution $p(\mathcal{A}|\mathcal{D})$ in the NAS search space. So, we approximate it with a variational distribution $p_{\boldsymbol{\alpha}}(\mathcal{A})$ (parameterized by a low-dimensional $\boldsymbol{\alpha}$) that can be optimized via variational inference, i.e., by minimizing the KL divergence between $p_{\boldsymbol{\alpha}}(\mathcal{A})$ and $p(\mathcal{A}|\mathcal{D})$. Equivalently, we only need to maximize a lower bound of the log-marginal likelihood (i.e., the *evidence lower bound* (ELBO) [Kingma and Welling, 2014]) to get an optimal variational distribution $p_{\boldsymbol{\alpha}^*}(\mathcal{A})$:

$$\max_{\boldsymbol{\alpha}} \mathbb{E}_{\mathcal{A} \sim p_{\boldsymbol{\alpha}}(\mathcal{A})} \left[\log p(\mathcal{D}|\mathcal{A})\right] - \text{KL}[p_{\boldsymbol{\alpha}}(\mathcal{A})||p(\mathcal{A})] . \quad (9)$$

Similar to [Kingma and Welling, 2014], a gradient-based optimization algorithm with the reparameterization trick is employed to solve (9) efficiently (see Appendix B.3). While Xie et al. [2019] have adopted a similar form to (9) (without the KL term) *during* the model training of the supernet (namely, the *best-response* posterior distribution), our *post-training* posterior distribution is able to not only provide a more accurate characterization of the single-model performances but also contribute to an improved ensemble search performance, as demonstrated in Appendix C.2.

## 3.3 BAYESIAN SAMPLING

To solve (6) effectively and efficiently, we finally introduce two novel Bayesian sampling algorithms based on the posterior distribution of architectures in Sec. 3.2, i.e., *Monte Carlo sampling* (MC Sampling) and *Stein Variational Gradient Descent with regularized diversity* (SVGD-RD), to

---

**Algorithm 1** NES via Bayesian Sampling (NESBS)

1: **Input:** Iterations $T$, ensemble size $n$, a supernet
2: Train the supernet to get its tuned parameters $\boldsymbol{\theta}^*$
3: Obtain the posterior distribution $p_{\boldsymbol{\alpha}^*}(\mathcal{A})$ with (9)
4: **for** iteration $t = 1, \dots, T$ **do**
5:     Sample $S_t$ of size $n$ via Algorithm 2 or 3
6:     Evaluate estimated $\mathcal{L}_{\text{val}}(\mathcal{F}_{S_t}(\boldsymbol{x}, \boldsymbol{\Theta}^*_{S_t}))$ given $\boldsymbol{\theta}^*$
7: **end for**
8: Select optimum $S^* = \arg\min_{S_t} \mathcal{L}_{\text{val}}(\mathcal{F}_{S_t}(\boldsymbol{x}, \boldsymbol{\Theta}^*_{S_t}))$

---

**Algorithm 2** MC Sampling

1: **Input:** Ensemble size $n$, set $S = \emptyset$, posterior $p_{\boldsymbol{\alpha}^*}(\mathcal{A})$
2: **for** iteration $i = 1, \dots, n$ **do**
3:     Sample $\mathcal{A}_i \sim p_{\boldsymbol{\alpha}^*}(\mathcal{A})$
4:     $S \leftarrow S \cup \{\mathcal{A}_i\}$
5: **end for**
6: **Output:** $S$

---

**Algorithm 3** SVGD-RD

1: **Input:** Diversity coefficient $\delta$, ensemble size $n$, iterations $L$, initial particles $\{\boldsymbol{x}_i^{(0)}\}_{i=1}^n$, posterior $p_{\boldsymbol{\alpha}^*}(\mathcal{A})$, kernel $k(\boldsymbol{x}, \boldsymbol{x}')$, step size $\{\epsilon_l\}_{l=1}^L$
2: **for** iteration $l = 0, \dots, L-1$ **do**
3:     Evaluate updates $\widehat{\phi}_l^*(\boldsymbol{x}) = \frac{1}{n}\sum_{j=1}^n \nabla_{\boldsymbol{x}_j^{(l)}} k(\boldsymbol{x}_j^{(l)}, \boldsymbol{x}) -$
    $\delta \nabla_{\boldsymbol{x}} k(\boldsymbol{x}_j^{(l)}, \boldsymbol{x}) + k(\boldsymbol{x}_j^{(l)}, \boldsymbol{x}) \nabla_{\boldsymbol{x}_j^{(l)}} \log p_{\boldsymbol{\alpha}^*}$
4:     Update particles $\boldsymbol{x}_i^{(l+1)} \leftarrow \boldsymbol{x}_i^{(l)} + \epsilon_l \, \widehat{\phi}_l^*(\boldsymbol{x}_i^{(l)})$
5: **end for**
6: **Output:** $S = \{\mathcal{A}_i\}_{i=1}^n$ derived based on $\{\boldsymbol{x}_i^{(L)}\}_{i=1}^n$

---

sample ensembles with both competitive single-model performances and compelling diversity of model predictions, as required by well-performing ensembles [Zhou, 2012].

### 3.3.1 Monte Carlo Sampling (MC Sampling)

Given the posterior distribution of architectures in Sec. 3.2, we firstly propose to use *Monte Carlo sampling* (MC Sampling) to sample a set of architectures from this posterior distribution (Algorithm 2). Note that MC Sampling guarantees that architectures with better single-model performances will be sampled (i.e., exploited) with higher probabilities, while architectures with diverse model predictions can also be sampled (i.e., explored) due to the inherent randomness in the sampling process. Compared with conventional NAS algorithms that select only one single well-performing architecture from the search space [Dong and Yang, 2019a, Xie et al., 2019], our MC sampling algorithm extends these algorithms by exploring the capability of diverse architectures while preserving its exploitation of architectures with

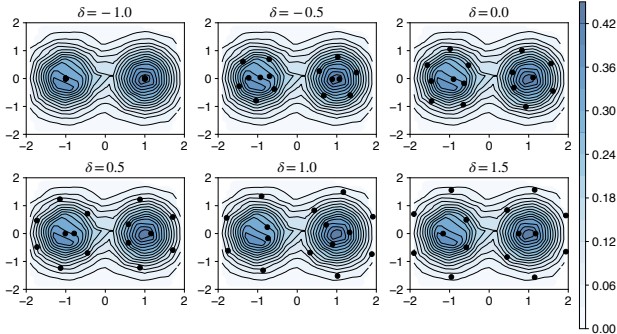

Figure 2: Impact of $\delta$ in SVGD-RD. We use contours and dots to denote the density of target distribution and sampled particles, respectively. The target distribution is chosen to be $p(\boldsymbol{x}) = (1/Z)\left[\mathcal{N}(\boldsymbol{x}|\boldsymbol{u}_1, \Sigma_1) + \mathcal{N}(\boldsymbol{x}|\boldsymbol{u}_2, \Sigma_2)\right]$ where $\boldsymbol{u}_1 = (-1, 0)$, $\boldsymbol{u}_2 = (0, 1)$, $\Sigma_1 = \Sigma_2 = \text{diag}((0.25, 0.5))$, and $Z$ denotes the normalization constant. Those sampled particles are obtained from Algorithm 3 using $L = 1000$, $n = 15$, $\epsilon_l = 0.1$, and a *radial basis function* (RBF) kernel. Notably, SVGD-RD tends to sample particles with more diverse probability densities as $\delta$ is increased, hence indicating a controllable (via $\delta$) diversity in our SVGD-RD algorithm. Meanwhile, SVGD-RD can consistently sample particles with high probability densities under varying $\delta$.

compelling single-model performances.

### 3.3.2 SVGD with Regularized Diversity (SVGD-RD)

However, the diversity of sampled architectures using the MC Sampling algorithm above cannot be controlled and hence may lead to poor ensemble search results. So, in order to achieve a controllable diversity, we resort to *Stein Variational Gradient Descent* (SVGD). Theoretically, SVGD is capable of sampling particles with both large probability density and good diversity where the diversity is explicitly encouraged (i.e., by the second term in (5)). Nonetheless, in practice, the particles sampled by SVGD may still fail to represent the target distribution well owing to the lack of diversity among those sampled particles, as observed in [Zhuo et al., 2018]. Besides, the diversity of sampled particles in standard SVGD still cannot be controlled by human experts.

We hence develop an *SVGD with regularized diversity* (SVGD-RD) sampling algorithm that can achieve a controllable diversity among those sampled particles. We follow the notations from Sec. 2.3. In particular, when optimizing the distribution $q^*$ (represented by the $n$ particles $\{\boldsymbol{x}_i^*\}_{i=1}^n$), we modify the objective in (1) by adding a term representing the (controllable) diversity among the particles measured by the kernel function $k(\boldsymbol{x}, \boldsymbol{x}')$:

$$q^* = \underset{q \in \mathcal{Q}}{\arg\min}\, \text{KL}(q\|p) + n\delta\, \mathbb{E}_{\boldsymbol{x}, \boldsymbol{x}' \sim q}\left[k(\boldsymbol{x}, \boldsymbol{x}')\right] \quad (10)$$

where $\delta$ is the parameter explicitly controlling the diversity, and $p$ in (10) denotes the posterior distribution $p_{\boldsymbol{\alpha}^*}(\mathcal{A})$ derived in Sec. 3.2 which we intend to sample from. Following the work of SVGD, $q^*$ in (10) is represented by $\{\boldsymbol{x}_i^*\}_{i=1}^n$ denoting our final selected neural network ensemble that can achieve both competitive single-model performances (i.e., large probability density) and also diverse model predictions. Proposition 1 below provides one possible update rule for the particles $\{\boldsymbol{x}_i\}_{i=1}^n$ to optimize (10) (see its proof in Appendix A). Finally, Algorithm 3 summarizes the details of our SVGD-RD algorithm and Appendix B.4 provides its optimization details in practice. After obtaining those optimal particles $\{\boldsymbol{x}_i^*\}_{i=1}^n$ in our SVGD-RD algorithm, we then apply these particles to derive the architectures in our final selected ensembles (see details in Appendix B.4).

**Proposition 1.** *Given the proximal operator* $\text{prox}_h(\boldsymbol{y}) = \arg\min_{\boldsymbol{z}} h(\boldsymbol{z}) + 1/2\|\boldsymbol{z} - \boldsymbol{y}\|_2^2$, *by applying proximal gradient method [Parikh and Boyd, 2014] and proper approximation,* (10) *can be optimized via the following updates of the particles* $\{\boldsymbol{x}_i\}_{i=1}^n$:

$$\boldsymbol{x}_i \leftarrow \boldsymbol{x}_i + \frac{1}{n}\sum_{j=1}^n k(\boldsymbol{x}_j, \boldsymbol{x}_i)\nabla_{\boldsymbol{x}_j}\log p(\boldsymbol{x}_j)$$
$$+ \nabla_{\boldsymbol{x}_j}k(\boldsymbol{x}_j, \boldsymbol{x}_i) - \delta\nabla_{\boldsymbol{x}_i}k(\boldsymbol{x}_j, \boldsymbol{x}_i)\,.$$

Compared with MC Sampling, our SVGD-RD algorithm provides a controllable trade-off between the single-model performances and the diverse model predictions. On the one hand, the minimization of the KL divergence term in (10) encourages the selection of architectures with competitive single-model performances by favoring particles with high probability densities, as shown by Proposition 2 below (its proof is in Appendix A).[2] On the other hand, the maximization of the scaled distance $-n\delta\, \mathbb{E}_{\boldsymbol{x}, \boldsymbol{x}' \sim q}\left[k(\boldsymbol{x}, \boldsymbol{x}')\right]$ among the sampled particles leads to a controllable diversity (via $\delta$) among these sampled particles and also a controllable diversity of the probability densities among these particles (see Fig. 2), which also implies a controllable diversity of the model predictions, as suggested in Sec. 3.2.

**Proposition 2.** *Let $p$ be a target density and $k(\boldsymbol{x}, \boldsymbol{x}') = c$ for every $\boldsymbol{x} = \boldsymbol{x}'$ where $c$ is a constant. For any $\delta \in \mathbb{R}$, our SVGD-RD algorithm is equivalent to the maximization of the density $p$ w.r.t. $\boldsymbol{x}$ in the case of $n = 1$.*

## 4 EXPERIMENTS

### 4.1 SEARCH IN NAS-BENCH-201

To verify the effectiveness and efficiency of our NESBS algorithm, we firstly compare it with other well-known

---

[2]Although Proposition 2 is only applicable in the case of $n = 1$, our SVGD-RD is still capable of sampling particles with high probability densities when $n > 1$, as validated in Fig. 2.

Table 1: Comparison of architectures selected by different NAS and ensemble (search) algorithms in NAS-Bench-201 with ensemble size $n = 3$. Test errors are reported with the mean and standard error of three independent trials and our search costs are evaluated on a single Nvidia 1080Ti GPU. Results marked by † are reported by Dong and Yang [2020].

| Architecture(s) | Test Error (%) | | | Search Cost (GPU Hours) |
|---|---|---|---|---|
| | **CIFAR-10** | **CIFAR-100** | **ImageNet-16-200** | |
| **Manual design** | | | | |
| ResNet† [He et al., 2016] | 6.03 | 29.14 | 56.37 | - |
| **NAS algorithms** | | | | |
| ENAS† [Pham et al., 2018] | 45.70±0.00 | 84.39±0.00 | 83.68±0.00 | 3.7 |
| DARTS† (2nd) [Liu et al., 2019] | 45.70±0.00 | 84.39±0.00 | 83.68±0.00 | 8.3 |
| GDAS† [Dong and Yang, 2019a] | 6.49±0.13 | 29.39±0.26 | 58.16±0.90 | 8.0 |
| SETN† [Dong and Yang, 2019b] | 13.81±4.63 | 43.13±7.77 | 68.10 ±4.07 | 8.6 |
| RSPS† [Li and Talwalkar, 2019] | 12.34±1.69 | 41.67±4.34 | 68.86±3.88 | 2.1 |
| **Ensemble (search) algorithms** | | | | |
| DeepEns [Lakshminarayanan et al., 2017] | 5.75 | 25.27 | 54.70 | - |
| NES-RS [Zaidi et al., 2021] | 5.83±0.33 | 25.58±0.84 | 54.34±1.67 | 5.1 |
| **Our ensemble search algorithm** | | | | |
| NESBS (MC Sampling) | 5.76±0.25 | 25.39±0.69 | **53.47**±1.75 | **1.1** |
| NESBS (SVGD-RD) | 5.92±0.07 | **25.00**±0.17 | **52.68**±0.35 | **1.2** |

NAS and ensemble (search) algorithms in NAS-Bench-201 [Dong and Yang, 2020]. Table 1 summarizes the results. Table 1 shows that ensemble (search) algorithms, including our NESBS, consistently achieve improved generalization performance over conventional NAS algorithms. This is because ensemble (search) algorithms will select neural network ensembles whereas NAS algorithms will select only one single architecture. Moreover, it has been widely verified that model ensembles generally outperform a single machine learning model in practice [Zhou, 2012]. In addition, our NESBS algorithm outperforms other ensemble (search) baseline (i.e., DeepEns and NES-RS), especially on large-scale datasets (i.e., CIFAR-100 [Krizhevsky, 2009] and ImageNet-16-200 [Chrabaszcz et al., 2017]) while incurring less search costs than NES-RS, which thus implies the superior performance of our NESBS over these ensemble (search) baselines. Even on a small-scale dataset (i.e., CIFAR-10), our NESBS can also achieve comparable search results to DeepEns and NES-RS. Interestingly, our NESBS algorithm is even able to incur reduced search costs than conventional NAS algorithms. This is likely because more training epochs have been used in these NAS algorithms, whereas a small number of training epochs can already contribute to well-performing results for our NESBS algorithm.

## 4.2 SEARCH IN THE DARTS SEARCH SPACE

We further demonstrate the superior search effectiveness and efficiency of our NESBS by comparing it with other NAS and ensemble (search) baselines in a larger search space (i.e., DARTS [Liu et al., 2019] search space) using both classification and adversarial defense tasks on CIFAR-10/100

or ImageNet [Deng et al., 2009]. We follow Appendix B.5 to evaluate the final neural network ensembles selected by our NESBS algorithm with ensemble size $n = 3$, $T = 5$, and optimization details in Appendix B.

**Ensemble for classification.** Table 2 summarizes the comparison of classification performances on CIFAR-10/100. Similar to the results in Sec. 4.1, ensemble (search) algorithms, including our NESBS, are generally able to achieve improved generalization performances over conventional NAS algorithms, which thus justifies the essence of ensemble (search) algorithms for improved performance. Notably, even compared with other ensemble baselines such as MC DropPath (i.e., developed following Monte Carlo Dropout [Gal and Ghahramani, 2016]) and DeepEns, our NESBS is still able to achieve improved performances. Since these ensemble baselines are orthogonal to our NESBS, they can be integrated into our NESBS for further performance improvement in real-world applications. More importantly, our algorithm outperforms NES-RS by achieving both improved search effectiveness (lowest test errors) and efficiency (lowest search costs). Furthermore, our NESBS even incurs comparable search costs compared with the most efficient NAS algorithms (e.g., GDAS, P-DARTS), which also highlights the efficiency of our NESBS. Similar results on ImageNet can be achieved by our NESBS as shown in Table 3. [3]

**Ensemble for adversarial defense.** Ensemble methods have already been shown to be an essential and effective de-

---

[3]Following the convention of NAS and ensemble search algorithms in Table 3, the ensembles selected by our NESBS are also searched on CIFAR-10 and then transferred to ImageNet.

Table 2: Comparison of different image classifiers on CIFAR-10/100. Results of MC DropPath are from a drop probability of 0.01 and our search costs are evaluated on Nvidia 1080Ti.

| Architecture(s) | Test Error (%) | | Params (M) | | Search Cost (GPU Days) | Search Method |
|---|---|---|---|---|---|---|
| | C10 | C100 | C10 | C100 | | |
| **NAS algorithms** | | | | | | |
| NASNet-A [Zoph et al., 2018] | 2.65 | - | 3.3 | - | 2000 | RL |
| AmoebaNet-A [Real et al., 2019] | 3.34 | 18.93 | 3.2 | 3.1 | 3150 | evolution |
| PNAS [Liu et al., 2018] | 3.41 | 19.53 | 3.2 | 3.2 | 225 | SMBO |
| ENAS [Pham et al., 2018] | 2.89 | 19.43 | 4.6 | 4.6 | 0.5 | RL |
| DARTS [Liu et al., 2019] | 2.76 | 17.54 | 3.3 | 3.4 | 1 | gradient |
| GDAS [Dong and Yang, 2019a] | 2.93 | 18.38 | 3.4 | 3.4 | 0.3 | gradient |
| P-DARTS [Chen et al., 2019] | 2.50 | - | 3.4 | - | 0.3 | gradient |
| DARTS- (avg) [Chu et al., 2020] | 2.59 | 17.51 | 3.5 | 3.3 | 0.4 | gradient |
| SDARTS-ADV [Chen and Hsieh, 2020] | 2.61 | - | 3.3 | - | 1.3 | gradient |
| **Ensemble (search) algorithms** | | | | | | |
| MC DropPath (ENAS) | 2.88 | 16.83 | 3.8‡ | 3.9‡ | - | - |
| DeepEns (ENAS) | 2.49 | 15.04 | 3.8‡ | 3.9‡ | - | - |
| DeepEns (DARTS) | 2.42 | 14.56 | 3.3‡ | 3.4‡ | - | - |
| NES-RS♯ [Zaidi et al., 2021] | 2.50 | 15.24 | 3.0‡ | 3.1‡ | 0.7 | greedy |
| **Our ensemble search algorithm** | | | | | | |
| NESBS (MC Sampling) | **2.41** | 14.70 | 3.8‡ | 3.9‡ | **0.2** | sampling |
| NESBS (SVGD-RD) | **2.36** | **14.55** | 3.7‡ | 3.8‡ | **0.2** | sampling |

‡ Reported as the averaged parameter size of the architectures in a neural network ensemble.
♯ Obtained from a pool of size 50, in which every architecture is uniformly randomly sampled from the DARTS search spaces and then trained independently for 50 epochs following the evaluation settings in Appendix B.5.

Table 3: Comparison of image classifiers on ImageNet. The ensemble size is set to $n = 3$ for NES-RS and NESBS.

| Architecture(s) | Test Error (%) | | Params (M) | +× (M) |
|---|---|---|---|---|
| | Top-1 | Top-5 | | |
| **NAS algorithms** | | | | |
| NASNet-A | 26.0 | 8.4 | 5.3 | 564 |
| AmoebaNet-A | 25.5 | 8.0 | 5.1 | 555 |
| PNAS | 25.8 | 8.1 | 5.1 | 588 |
| DARTS | 26.7 | 8.7 | 4.7 | 574 |
| GDAS | 26.0 | 8.5 | 5.3 | 581 |
| P-DARTS | 24.4 | 7.4 | 4.9 | 557 |
| SDARTS-ADV | 25.2 | 7.8 | 5.4 | 594 |
| **Ensemble (search) algorithm** | | | | |
| NES-RS | 23.4 | 6.8 | 3.9 | 432 |
| **Our ensemble search algorithm** | | | | |
| NESBS (MC Sampling) | **22.3** | **6.2** | 4.6 | 522 |
| NESBS (SVGD-RD) | **22.3** | **6.1** | 4.9 | 562 |

fense mechanism against adversarial attacks [Strauss et al., 2017]. Specifically, an adversarial attacker can only use *a single model* randomly sampled from an ensemble to generate the adversarial examples, whereas the ensemble method defends against adversarial attacks (i.e., makes its predictions) using *all models* in this ensemble. Ensemble methods

can defend against the adversarial attacks in such a setting because the generated adversarial examples using only one single model are unlikely to fool all models in an ensemble. More details are provided in Appendix B.6. Table 4 summarizes the comparison of adversarial defense among ensemble (search) algorithms on CIFAR-10/100 under different white-box adversarial attacks, including the *Fast Gradient Signed Method* (FGSM) attack Goodfellow et al. [2015], the *Projected Gradient Descent* (PGD) attack Madry et al. [2018], the *Carlini Wagner* (CW) attack Carlini and Wagner [2017], and the AutoAttack [Croce and Hein, 2020]. Table 4 shows that ensemble (search) algorithms are indeed able to significantly improve the performance of adversarial defense, i.e., the test accuracies in the *Defense* column are consistently higher than the ones in *Attack* column. More importantly, even under different white-box adversarial attacks, our NESBS algorithm can generally achieve improved defense performances (i.e., higher test accuracy in the *Defense* columns) than other baselines including DeepEns and NES-RS. These results thus further support the effectiveness of our NESBS over existing ensemble (search) algorithms. Besides, even regarding the adversarial robustness of the single models in an ensemble, the architectures selected by our NESBS are also more advanced (i.e., by achieving higher test accuracy in the *Attack* columns) than well-known architectures such as RobNet [Guo et al., 2020] and DARTS.

Table 4: Comparison of adversarial defense among different ensemble (search) algorithms on CIFAR-10/100 under white-box adversarial attacks. The *Attack* and *Defense* columns denote the test *accuracy* under the attack using a single model randomly sampled from an ensemble and the defense using the whole ensemble, respectively. Each result reports the mean and standard deviation of test accuracies for 3 rounds of the attack-defense process with an ensemble size of $n = 3$.

| Method | FGSM | | PGD-40 | | CW | | AutoAttack | |
|---|---|---|---|---|---|---|---|---|
| | Attack (%) | Defense (%) | Attack (%) | Defense (%) | Attack (%) | Defense (%) | Attack (%) | Defense (%) |
| **On CIFAR-10 Dataset** | | | | | | | | |
| DeepEns | - | - | - | - | - | - | - | - |
| ↪ RobNet-free | 66.62±0.32 | 85.25±0.39 | 41.81±0.80 | 77.48±0.67 | 5.74±1.41 | 86.53±0.50 | 21.35±0.33 | 45.51±0.15 |
| ↪ ENAS | 77.85±0.58 | 87.94±0.21 | 59.51±1.13 | 86.57±0.15 | 31.36±1.20 | 85.20±0.77 | 31.71±0.72 | 50.96±0.07 |
| ↪ DARTS | 76.79±0.80 | 88.21±0.14 | 57.71±1.65 | 82.02±0.10 | 26.90±1.37 | 82.46±0.35 | 29.97±1.17 | 49.67±0.14 |
| NES-RS | 79.19±1.39 | 89.32±0.27 | 65.59±2.11 | 85.22±0.41 | 37.20±4.62 | 86.75±0.88 | 35.00±1.15 | 53.80±0.14 |
| NESBS (MC Sampling) | 78.75±1.29 | 89.15±0.08 | 63.60±1.87 | 85.35±0.31 | **37.71**±1.97 | **86.86**±0.66 | 36.02±0.64 | **56.90**±0.17 |
| NESBS (SVGD-RD) | 79.12±0.61 | **89.86**±0.33 | 65.53±1.56 | 85.37±0.38 | **38.27**±1.27 | 86.00±1.10 | **37.55**±0.68 | **57.15**±0.20 |
| **On CIFAR-100 Dataset** | | | | | | | | |
| DeepEns | - | - | - | - | - | - | - | - |
| ↪ RobNet-free | 36.47±0.25 | 61.39±0.30 | 18.18±0.47 | 52.61±0.13 | 2.36±0.13 | 69.44±0.04 | 7.31±0.35 | 24.56±0.33 |
| ↪ ENAS | 46.40±0.37 | 64.94±0.27 | 28.87±0.27 | 56.79±0.25 | 9.60±0.30 | 69.43±0.44 | 11.53±0.47 | 27.01±0.27 |
| ↪ DARTS | 46.98±0.57 | 65.38±0.23 | 28.78±0.74 | 57.10±0.04 | 9.73±0.43 | 70.15±0.29 | 11.20±0.40 | 26.86±0.36 |
| NES-RS | 47.10±1.46 | 65.33±0.36 | 30.68±1.66 | 58.80±0.80 | 9.96±1.45 | 70.24±0.33 | 12.01±0.93 | 27.49±0.34 |
| NESBS (MC Sampling) | **50.69**±1.58 | **67.63**±0.05 | 33.37±0.42 | **60.36**±0.62 | 15.64±2.83 | **71.25**±1.27 | 13.11±1.16 | 29.87±1.17 |
| NESBS (SVGD-RD) | **51.47**±0.40 | 66.66±0.13 | **35.02**±0.37 | 59.96±0.18 | **16.72**±0.61 | 69.88±0.16 | **14.62**±0.55 | **31.07**±0.33 |

## 4.3 SINGLE-MODEL PERFORMANCES AND DIVERSE MODEL PREDICTIONS

We demonstrate that the effectiveness of our NESBS results from its ability to achieve a good trade-off between the single-model performances and the diversity of model predictions. We firstly quantitatively compare the single-model performances (measured by the *averaged test error* (ATE) of the models in an ensemble) and the diversity of model predictions (measured by the *pairwise predictive disagreement* (PPD) of an ensemble [Fort et al., 2019]) achieved by different ensemble (search) algorithms on CIFAR-10/100. We further qualitatively visualize their single-model performances and diverse model predictions using a histogram of the ATE of the models in their ensembles and a t-SNE [van der Maaten and Hinton, 2008] plot of their model predictions, respectively.

Table 5 and Fig. 3 present the results of our quantitative and qualitative comparisons, respectively. Compared with the ensemble baselines of MC DropPath and DeepEns, our NESBS is capable of enjoying a larger diversity of model predictions while preserving competitive single-model performances. Meanwhile, compared with the ensemble search baselines of NES-RS, our algorithm can achieve improved single-model performances while maintaining comparably diverse model predictions. These results suggest that our NESBS is able to select ensembles achieving a better trade-off between the single-model performances and the diversity of model predictions among these baselines, which is known to be an important criterion for well-performing ensembles [Zhou, 2012]. Thus, Table 5 and Fig. 3 provide empirical justifications for the improved effectiveness of NESBS.

Table 5: Quantitative comparison of the single-model performances (measured by ATE (%), smaller is better) and the diversity of model predictions (measured by PPD (%), larger is better) achieved by different ensemble (search) algorithms with an ensemble size of 3 on CIFAR-10/100.

| Method | C10 | | C100 | |
|---|---|---|---|---|
| | ATE | PPD | ATE | PPD |
| MC DropPath (DARTS) | 2.71 | 0.39 | 16.68 | 2.63 |
| DeepEns (DARTS) | **2.69** | 2.08 | **16.18** | 12.45 |
| NES-RS | 2.87 | 2.29 | 17.20 | **14.14** |
| NESBS (MC Sampling) | 2.80 | **2.57** | 16.70 | 13.84 |
| NESBS (SVGD-RD) | 2.78 | 2.27 | 16.50 | 13.16 |

## 5 CONCLUSION

This paper presents a novel neural ensemble search algorithms, called NESBS, that can effectively and efficiently select well-performing neural network ensembles with diverse architectures from a NAS search space. Our extensive experiments have shown that NESBS is able to achieve improved performances while preserving a comparable search cost compared with conventional NAS algorithms. Moreover, even compared with other ensemble (search) baselines (e.g., DeepEns and NES-RS), our NESBS is also capable of enjoying boosted search effectiveness and efficiency, which further suggests the superior performance of our NESBS in practice.

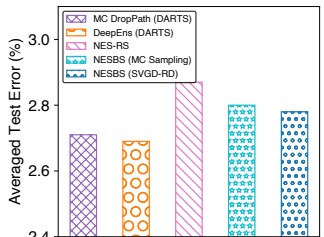 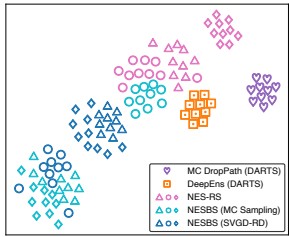

(a) Single-model performances   (b) Diverse predictions

Figure 3: Qualitative comparison of (a) the single-model performances and (b) the diverse model predictions achieved by different ensemble (search) algorithms with an ensemble size of $n = 3$ on CIFAR-10. Each architecture in (b) is independently evaluated for ten times to visualize their model predictions, which follows from DeepEns.

## Acknowledgements

This research/project is supported by A*STAR under its RIE2020 Advanced Manufacturing and Engineering (AME) Programmatic Funds (Award A20H6b0151).

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
