# OpenReview forum: "Neural Ensemble Search via Bayesian Sampling"
_auai.org/UAI/2022/Conference — UAI 2022 Poster_

### Official Review · Reviewer_9jsi · 2022-04-10

**Q2(1) Originality/Novelty:** 3
**Q2(2) Significance/Impact:** 3
**Q2(3) Correctness/Technical Quality:** 3
**Q2(6) Clarity Of Writing:** 4
**Q6 Overall Score:** 7
**Q8 Confidence In Your Score:** 1

**Q1 Summary And Contributions:**

The authors propose a neural ensemble search algorithm via Bayesian sampling.

**Q2 Assessment Of The Paper:**

More detailed information regarding each of these aspects is given below:

**Q2(4) Quality Of Experiments (Optional):**

3: Good: The experimental evaluation is adequate, and the results convincingly support the main claims.

**Q2(5) Reproducibility:**

4: Excellent: Key resources (e.g., proofs, code, data) are available and key details (e.g., proof sketches, experimental setup) are comprehensively described for competent researchers to confidently and easily reproduce the main results.

**Q3 Main Strengths:**

The paper is well written, and it tackles an interesting problem with what appear to be good results.

**Q4 Main Weakness:**

Nothing specific, it appears a solid paper.

**Q5 Detailed Comments To The Authors:**

The paper is well written, and it tackles an interesting problem with what appear to be good results.
I believe the results will be of great interest, at least to a sub-area of the UAI community.
The authors provided their source code in the supplementary material. Hence I am confident they'll release it upon acceptance.

Just a minor typo:
*Monto* Carlo Dropout -> at page 2

**Q7 Justification For Your Score:**

This appears to be a technically solid paper with impact in at least a sub-area of the UAI community. The evaluation looks very good, with no issue concerning reproducibility and ethics.

**Q9 Complying With Reviewing Instructions:**

1: Yes.

---

### Official Review · Reviewer_w1DB · 2022-04-12

**Q2(1) Originality/Novelty:** 3
**Q2(2) Significance/Impact:** 2
**Q2(3) Correctness/Technical Quality:** 2
**Q2(6) Clarity Of Writing:** 3
**Q6 Overall Score:** 5
**Q8 Confidence In Your Score:** 4

**Q1 Summary And Contributions:**

This paper claimed that current Neural Architecture Search (NAS) methods could just select one single well-performing architecture. To overcome it, they introduce a novel neural ensemble search algorithm, called neural ensemble search via Bayesian sampling (NESBS), which could choose the well-performing neural network ensemble with diverse architectures from the search space.

**Q2 Assessment Of The Paper:**

More detailed information regarding each of these aspects is given below:

**Q2(4) Quality Of Experiments (Optional):**

3: Good: The experimental evaluation is adequate, and the results convincingly support the main claims.

**Q2(5) Reproducibility:**

2: Fair: Key resources (e.g., proofs, code, data) are unavailable but key details (e.g., proof sketches, experimental setup) are sufficiently well-described for an expert to confidently reproduce the main results.

**Q3 Main Strengths:**

The whole paper gives a comprehensive introduction to NESBS. For Section 3, this paper give a rational explication of the theory and effect of each sub-algorithm. Additionally, Figures 1-3 make the expression idea very straightforward. For the performance of the NESBS method, from the experimental section, it attain effectively and efficiently better results compared with some baselines. Additionally, in the adversarial defense case, NESBS also expresses better robustness.

**Q4 Main Weakness:**

According to Section 3, Monte Carlo Sampling (MC Sampling) and SVGD with Regularized Diversity (SVGD-RD) as two Bayesian sampling methods are the main points that could make the final NESBS more effective and efficient. However, these two sampling methods are mutually independent and the final performance of these two methods in Section 4 are just similar. Therefore, how to choose the specific Bayesian sampling method when adopting NESBS?

**Q5 Detailed Comments To The Authors:**

The whole paper comprehensively express the main idea of the NESBS. However, some detailed problems need to be revised.
1) Line 3 in Algorithm 3 needs to be adjusted.
2) The position of caption Figure 3 (a) and Figure 3 (b) overlap.
3) How to measure GPU hours in Table 1.

**Q7 Justification For Your Score:**

In a word, this paper give a sufficient presentation about their NESBS from the methodology and experimental section but only some minor problems need to be addressed.  However, the design of NESBS is straightforward and mainly refers to the currently existing methods. Therefore, I will give the corresponding score in Q6.

**Q9 Complying With Reviewing Instructions:**

1: Yes.

---

### Official Review · Reviewer_6fyP · 2022-04-27

**Q2(1) Originality/Novelty:** 2
**Q2(2) Significance/Impact:** 3
**Q2(3) Correctness/Technical Quality:** 2
**Q2(6) Clarity Of Writing:** 2
**Q6 Overall Score:** 6
**Q8 Confidence In Your Score:** 3

**Q1 Summary And Contributions:**

This paper introduced neural ensemble search via Bayesian sampling (NESBS), a novel neural ensemble search algorithm for selecting well-performing neural network ensembles from a NAS search space. NESBS algorithm is shown to be able to achieve improved performance over state-of-the-art NAS algorithms while incurring a comparable search cost, indicating the superior of our NESBS algorithm over these conventional NAS algorithms in practice.

**Q2 Assessment Of The Paper:**

More detailed information regarding each of these aspects is given below:

**Q2(4) Quality Of Experiments (Optional):**

3: Good: The experimental evaluation is adequate, and the results convincingly support the main claims.

**Q2(5) Reproducibility:**

3: Good: Key resources (e.g., proofs, code, data) are available and key details (e.g., proofs, experimental setup) are sufficiently well-described for competent researchers to confidently reproduce the main results.

**Q3 Main Strengths:**

+ The usage of Bayesian sampling for neural ensemble search seems novel

+ The experiments are relatively thorough, covering three datasets

**Q4 Main Weakness:**

- There are already methods using NAS for adversarial defense, e.g., see [1] and others. In "Ensemble for adversarial defense" section, however, the authors failed to compare NESBS with these existing methods, making the results less convincing.

[1] Guo et al. When nas meets robustness: In search of robust architectures against adversarial attacks. CVPR 2020.

- The adversarial attacks used are also outdated and less effective. At least run Autoattack [2] to see whether the robust accuracy is actually converged.

[2] Reliable evaluation of adversarial robustness with an ensemble of diverse parameter-free attacks. ICML 2020

- Another issue is that all of the tested datasets are of low resolution (CIFAR  and downsampled ImageNet). Does the method work on practical images with higher resolution?

**Q5 Detailed Comments To The Authors:**

Please see the main weaknesses part. The results for adversarial defense are not well justified as it lacks comparison with existing NAS-based methods.

More advanced attacks should be tested to verify the effectiveness.

**Q7 Justification For Your Score:**

Currently the weaknesses outweigh strengths. I would like to see the author response on my points.

**Q9 Complying With Reviewing Instructions:**

1: Yes.

---

### Official Review · Reviewer_pifW · 2022-04-27

**Q2(1) Originality/Novelty:** 2
**Q2(2) Significance/Impact:** 2
**Q2(3) Correctness/Technical Quality:** 3
**Q2(6) Clarity Of Writing:** 4
**Q6 Overall Score:** 6
**Q8 Confidence In Your Score:** 4

**Q1 Summary And Contributions:**

The paper proposes an algorithm for searching/proposing ensembles of neural networks for an specific task. The algorithm is achieving competitive results while keeping the search time reasonable.

**Q2 Assessment Of The Paper:**

More detailed information regarding each of these aspects is given below:

**Q2(4) Quality Of Experiments (Optional):**

3: Good: The experimental evaluation is adequate, and the results convincingly support the main claims.

**Q2(5) Reproducibility:**

3: Good: Key resources (e.g., proofs, code, data) are available and key details (e.g., proofs, experimental setup) are sufficiently well-described for competent researchers to confidently reproduce the main results.

**Q3 Main Strengths:**

The algorithm seems to be simple, and reasonable. The experiments are well designed, and they cover a nice range of use cases. It seems to have a nice search cost score. The supplementary material nicely extends the knowledge about the proposed idea.

The paper is well written. It has nice figures, plots and tables that make it easy to understand.

**Q4 Main Weakness:**

Despite I find the paper quite reasonable, I am afraid the impact of the paper is low. I find it like another approach to perform NES in which the performance seems to be marginally better. The experiments are the average of just a few repetitions, and by looking at the difference in the means, and the variances, I think it would be difficult to say whether the proposed algorithm is clearly better. My criticism here is not about the results itself, but the impact that the work might have based on them. Maybe publishing a notebook could help with the impact of this work.

I would like to have a discussion on the hyper-parameters of the algorithm. For example, I have the feeling that the Diversity Coefficient needs to be carefully chosen. I would suggest to move the adversarial attack experiment to the appendix. I find the robustness against adversarial attacks is a feature of the NES models, rather than a problem that your idea is trying to solve.

**Q5 Detailed Comments To The Authors:**

I think the paper looks nice, but in my opinion it could be a bit improved. You can take a look to the Q4.

I also would like to rise the score for the impact if the code to replicate at least one experiment is release. I think if somebody else can replicate the results or adapt them is a nice way to justify a greater impact.


**Q7 Justification For Your Score:**

My main concerns for the score are the impact of the work, and the discussion of the hyper-parameters that are needed should be included in the main paper. You can take a look to Q4 for a longer explanation.

**Q9 Complying With Reviewing Instructions:**

1: Yes.

---

### Decision · Program_Chairs · 2022-05-15

**Decision:**

Accept (Poster)

**Comment:**

Meta Review: This work aims to find an ensemble of neural networks which are robust to adversarial attacks.

The quality of the paper is appropriate for acceptance. While the reviewers all agree that the originality of the work is somewhat limited, the proposed approach is still somewhat novel. The experiments are well-designed and demonstrate the robustness of the approach to adversarial approaches. The paper is clearly written. Taken together, it seems the paper could be modestly significant.

Pros
* Simple, reasonable Bayesian sampling algorithm
* Well-designed experiments
* Well-written
* Experiments show robustness to adversarial sampling

Cons
* Limited novelty
* Limited performance improvement relative to SOTA
* Choice of Bayesian sampling algorithm does not matter
* Adversarial attacks are outdated